# Outcomes of complex decongestive therapy in managing upper limb lymphedema in female breast cancer patients at a palliative care unit of a tertiary care hospital in Bangladesh

Rifat Jahan[1], A.K.M. Motiur Rahman Bhuiyan[1], Afroja Alam[1]*,
Mostofa Kamal Chowdhury[1], Romena Hassan[2], Nisha Mortuja Aktar[3]

1 Bangladesh Medical University (BMU), Dhaka, Bangladesh, 2 New Cross Hospital, The Royal Wolverhampton Trust, United Kingdom, 3 National Institute of Cancer Research & Hospital (NICRH), Dhaka, Bangladesh

* afrojaalamk55@gmail.com

## Abstract

### Background

Lymphedema is a chronic condition that significantly affects both physical function and quality of life of breast cancer patients. Although there is no definitive cure, various treatment options exist to alleviate its symptoms. Among these, Complex Decongestive Therapy (CDT) is widely regarded as a primary approach. This study seeks to evaluate the effectiveness of CDT for breast cancer patients with upper limb lymphedema and aims to assess the benefits of this treatment despite the challenges and constraints in resource-limited settings.

### Methods

This observational study was conducted among 42 female breast cancer patients with unilateral upper limb lymphedema attending the Lymphedema Clinic of the Department of Palliative Medicine at Bangladesh Medical University in Dhaka, Bangladesh. Limb volume, skin condition, and clinical signs and symptoms were assessed at baseline. All patients received the intensive phase of Complex Decongestive Therapy (CDT) for 6 weeks, with follow-up assessments conducted at the 3rd week and the 6th week.

### Result

A significant reduction in the volume of the affected limbs was observed from baseline to the 6th week, as well as from the 3rd week to the 6th week. Although no statistically significant improvement in skin edema was recorded during this period, visible clinical improvement in skin texture was noted. After receiving CDT and proper skin

**Data availability statement:** All data relevant to the study are accessible in figshare doi: 10.6084/m9.figshare.28906847

**Funding:** This study is funded by Bangladesh Medical University, Dhaka, Bangladesh, Grant no. 136, awarded to RJ. The funders played no role in the study design, data collection and analysis, decision to publish, or preparation of the manuscript.

**Competing interests:** The authors have declared that no competing interests exist.

care, 59.5% of patients regained normal skin on the affected limb. Additionally, there was a significant reduction in self-reported symptoms such as tightness, heaviness, and pain in the affected limb from baseline to the 6th week.

## Conclusion

Lymphedema management using all components of Complex Decongestive Therapy (CDT) was found to be effective in reducing limb volume and alleviating the distressing symptoms of patients. Timely referral of lymphedema patients to specialized clinics and initiation of CDT can significantly reduce their ongoing suffering in Bangladesh.

## Background

Lymphedema is the abnormal accumulation of lymphatic fluid in the interstitial tissue, characterized by high protein content, due to obstruction or compression in lymphatic outflow [1]. It can occur anywhere in the body; however, upper limb lymphedema is most commonly associated with breast malignancy, often as a symptom of advanced cancer or squeale of anti-cancer treatments [2].

Approximately 12% to 60% of lymphedema cases have been reported in breast cancer patients, with around 27% presenting with upper limb lymphedema [3]. In these patients, this condition is typically caused by surgical removal of axillary lymph nodes, radiotherapy, tumor obstruction, or a combination of these factors. It may also result from postoperative infections or lymph node metastasis [2,4].

Common symptoms include tightness, heaviness of the affected limb, a bursting sensation, pain, functional impairment, restricted mobility, and difficulty finding well-fitting clothing or footwear. Complications may include lymphangiectasis, hyperkeratosis, papillomatosis, cutaneous fibrosis, and lymphorrhea [5]. Lymphedema can further increase the risk of infections, reduce range of motion, alter sensory perception, lower self-esteem, and lead to greater dependency, psychological distress, and social isolation [6,7]. If left untreated, it can significantly impair a patient's quality of life [8].

Early detection through proper assessment, followed by timely and appropriate treatment, is essential to prevent the progression of this debilitating condition. In palliative care settings, the primary management approach for lymphedema is conservative therapy, with Complex Decongestive Therapy (CDT) recognized as the gold standard. CDT primarily consists of compression (via multilayer bandaging or compression garments), manual lymphatic drainage (MLD), skin care, and therapeutic exercise. It remains the most effective and widely adopted method for managing lymphedema [9].

Complex Decongestive Therapy (CDT) is typically administered in two phases. Intensive CDT, or Phase 1, is recommended for patients with moderate to severe lymphedema and includes manual lymphatic drainage (MLD), compression bandaging (CB), therapeutic exercise, and skin care. Modified CDT, or Phase 2, is generally used for mild to moderate cases and focuses on maintenance through compression

garments or bandaging, continued skin care, and additional MLD or exercise as needed [10,11]. These four core components are adapted based on patient needs and tolerance. When administered appropriately by trained professionals, CDT can achieve a limb volume reduction of 50% to 70%, with results sustained over time [11].

In Bangladesh, palliative care services remain limited, ranking among the lowest globally in terms of availability and accessibility, despite a high burden of unmet palliative care needs. Since 2007, the Centre for Palliative Care at Bangladesh Medical University (BMU) has been at the forefront of palliative care development in the country. As part of its services, the university established a dedicated lymphedema clinic that operates weekly, providing much needed care to patients with lymphedema.

However, studies on lymphedema management in palliative care patients in Bangladesh remain scarce. In our experience, delayed presentation, limited resources, inadequate healthcare facilities, and restricted treatment options pose significant challenges in the effective management of lymphedema symptoms, particularly among breast cancer patients. In this context, the present study aims to evaluate the effectiveness of Complex Decongestive Therapy (CDT) in managing lymphedema among this patient population and to explore the therapeutic benefits of CDT despite existing systemic limitations.

## Methods

### Study design and setting

This observational study was conducted at the Lymphedema Clinic of the Department of Palliative Medicine, Bangladesh Medical University (BMU), Shahbagh, Dhaka. Participant recruitment commenced on 1 May 2023 and continued through 18 August 2023. The six-week post-treatment follow-up for all enrolled patients was concluded by 30 September 2023.

### Sample criteria

All female breast cancer patients with moderate to massive unilateral upper limb lymphedema attending the Lymphedema Clinic for their first session of CDT were included in the study. Patients with lower limb lymphedema or non-cancer-related lymphedema, those who received any treatments that might reduce lymphedema (such as previous sessions of CDT, radiotherapy or diuretics), those with organ failure (heart, liver, or renal), or those with contraindications for CDT (recent deep vein thrombosis, cellulitis, or total blockage by a tumor) were excluded.

### Sample size

The sample was selected using the census method. During the study period, a total of 51 patients attended the Lymphedema Clinic for first session of CDT. However, 9 patients were lost during follow-up, leaving a final sample size of 42.

### Lymphedema grading

The severity of lymphedema in the affected limb was categorized based on circumferential measurements in comparison to the contralateral, unaffected limb [12]-

- Grade 1 (Mild): Involves distal regions such as the forearm and hand or lower leg and foot. Circumferential difference is less than 4 cm, with no observable tissue changes.

- Grade 2 (Moderate): Affects the entire limb or the corresponding trunk quadrant, with a circumferential difference of 4–6 cm. Observable tissue alterations such as pitting may occur, and patients may present with erysipelas.

- Grade 3a (Severe): Lymphedema extends to one limb and its adjacent trunk quadrant, with a circumferential difference exceeding 6 cm. Skin changes including cornification, keratosis, cysts, or fistulae are evident, and recurrent episodes of erysipelas may be reported.

- Grade 3b (Massive): Characterized by the involvement of two or more extremities, exhibiting the same features as Grade 3a.

- Grade 4 (Gigantic): Marked by extreme enlargement of the affected limbs, attributed to near-total obstruction of lymphatic flow.

## Complex decompression therapy (CDT) procedure

Complex decongestive therapy (CDT) is a physiotherapeutic intervention composed of several components, including manual lymphatic drainage (MLD), compression bandages or garments, exercise, and skin care. Intensive CDT was given to moderate to severe lymphedema, which continues until there is significant volume reduction of the limb [10].

Manual lymphatic drainage (MLD) is a specialized form of massage used by lymphedema specialists to guide fluid from swollen areas to regions where lymphatics are functioning normally. In advanced stages of the disease, MLD may not be clinically effective or indicated. However, it can be modified to provide a soothing effect for patients and offer caregivers a sense of relief [4].

Compression therapy consists of multilayer bandaging and compression garments or hosiery. Multilayer bandages are applied during the intensive phase to reduce volume, while compression garments are used during the maintenance phase. The compression garment should be worn throughout the day and removed at night, helping to prevent further swelling [13].

Exercise is encouraged in these patients to support lymphatic flow, as muscle contraction helps enhance lymphatic circulation through the vessels. In cases of advanced disease, passive movement can be performed by slowly moving the joints. In advanced cancer, exercise typically helps maintain function rather than improve it [5].

Skin care for lymphedema includes moisturizing, maintaining cleanliness, preventing cuts and bites, and educating patients [5].

## Limb volume measurement technique

Limb volume changes were measured by assessing the circumference of the affected limb using a measuring tape at fixed designated points, and the limb volume was then calculated using the truncated cone formula [14].

Volume of a segment, $V = h \times (C^2 + Cc + c^2)/(\pi \times 12)$
- h = segment length (between 2 measurement of points)
- C and c are circumferences at each end of segment
- Difference in volume (ml): (final volume-initial volume) ÷ initial volume

*Measurement points:* Arm: knuckles, styloid process, elbow crease, 10 cm above elbow [15].

## Skin integrity assessment technique

Skin quality of the affected limb were assessed in two domains [16]-

Presence of edema: pitting, non-pitting, no edema

Texture: Normal, dry, shiny, hyperkeratosis, lymphoceles, papillomatosis, lymphorrhea

## Data collection procedure

The research team visited the Lymphedema Clinic every Monday. The study's purpose was explained to the patients, and written informed consent was obtained. Before starting the CDT procedure, volume measurement, lymphedema grading, and assessment of skin integrity of the affected limb were performed according to the aforementioned technique. Sociodemographic and clinical characteristics of the patients were also recorded on a data collection sheet. The CDT procedure was then applied by the attending physicians and trained palliative care assistants at the clinic. All patients underwent

intensive CDT phase 1 for 6 weeks during the study period. No alterations to the treatment protocol were made by the investigators. The patients were followed up after 3 weeks during the mid-procedure and after 6 weeks after competition of 6 weeks of CDT. During both follow-up sessions, clinical sign and symptoms, volume measurements and skin integrity of the affected limb were reassessed.

### Data analysis

Data analysis was performed using the Statistical Package for the Social Sciences (SPSS), version 26. Categorical data, such as age, educational status, site of metastasis, current symptoms, and mode of anticancer treatment, were expressed as frequencies and percentages. Volume changes and skin integrity were measured at baseline and during the 1st and 2nd follow-up visits. To compare the mean volume changes between visits, the Bonferroni test was used. One-way ANOVA was applied to assess associations among the three groups. Skin condition and symptom changes were assessed by Chi-squre test. All means were calculated with a 95% confidence interval, and p-values < 0.05 were considered statistically significant.

### Ethical considerations

The ethical approval (Approval no: BSMMU/2023/2546, Date: 05/03/2023) was obtained from the Institutional Review Board (IRB) of Bangladesh Medical University, Bangladesh. Written informed consent was taken from all the eligible patients.

## Results

The mean age of the participants was 50.2 ± 8.6 years. The majority of patients (54.8%) were diagnosed at stage III cancer. Nearly half of the patients (47.6%) were diagnosed with lymph node (LN) metastasis during the study period, and the majority (71.4%) were symptomatic but fully ambulant. All patients presented with unilateral lymphedema, with the left upper limb being the most frequently affected site (54.8%). More than half (66.6%) patients had Grade 2 lymphedema. Almost all the patients (92.9%) underwent surgery, followed by chemotherapy in 90.5% and radiotherapy in 76.2%. Mean duration of development of lymphedema from last treatment was 16.6 ± 10 months (Table 1).
After receiving CDT, the volume of the affected upper limbs in patients decreased from baseline to subsequent follow-up visits (Table 2).

Significant volume reduction in both the right and left upper limbs was observed from baseline to the 2nd follow-up visit, as well as from the 1st to the 2nd follow-up visit. However, no significant volume change was noted from baseline to the 1st follow-up visit. This indicates that CDT effectively reduced limb volume over the study period. However, we did not find any association between cancer stage and the volume reduction of the affected limb (Table 3).

No statistically significant improvement in skin edema was observed between the baseline visit and the 2nd follow-up visit. However, visible clinical improvement in the skin texture was noted. Approximately 76.2% of patients presented with dry, shiny skin on the affected limb at baseline. After receiving CDT and proper skin care, 59.5% of patients regained normal skin on the affected limb, although this improvement was not statistically significant. However, the improvement in skin condition can be attributed to CDT, as it not only resulted in volume reduction but also included proper skin care (Table 4).
Most patients (92.9%) initially reported a sensation of heaviness, while 90.5% experienced tightness in the affected limb, along with difficulties in finding well-fitting clothes. However, following subsequent visits, there was a significant reduction in tightness, heaviness, and pain in the affected limb (Table 5).

## Discussion

Lymphedema is a chronic condition that markedly impairs both physical function and quality of life of the patients. Although no definitive cure currently exists, various treatment modalities are available to mitigate its impact. Among these, Complex Decongestive Therapy (CDT) is widely regarded as the first-line treatment approach.

**Table 1. Distribution of the participants according to socio-demographic characteristics (n = 42).**

| Variables | Frequency (n) | Percentage (%) |
|---|---|---|
| **Age** | | |
| Mean±SD | 50.2±8.6 | |
| **Stage of breast cancer** | | |
| Stage I | 3 | 7.1 |
| Stage II | 16 | 38.1 |
| Stage III | 23 | 54.8 |
| **Affected limb** | | |
| Left upper limb | 23 | 54.8 |
| Right upper limb | 19 | 45.2 |
| **Site of metastasis** | | |
| Multiple metastasis | 1 | 2.3 |
| Bone | 2 | 4.8 |
| Opposite breast | 1 | 2.3 |
| Lymph nodes | 20 | 47.6 |
| No metastasis | 18 | 42.6 |
| **Grade of lymphedema** | | |
| Grade 2 | 28 | 66.6 |
| Grade 3a | 14 | 33.4 |
| **Performance status** | | |
| Symptomatic, but completely ambulant | 30 | 71.4 |
| Symptomatic, spends <50% time in bed during the day | 12 | 28.6 |
| **Anticancer treatment*** | | |
| Surgery | 39 | 92.9 |
| Chemotherapy | 38 | 90.5 |
| Radiotherapy | 32 | 76.2 |
| Hormone therapy | 4 | 9.5 |
| Immunotherapy | 24 | 57.1 |
| **Mean duration of lymphedema development** | 16.6±10 months | |

*Multiple response

**Table 2. Association of volume changes of lymphedema limbs among different visits (n = 42).**

| Parameters | Baseline | 1st follow-up visit | 2nd follow-up visit | p value* |
|---|---|---|---|---|
| **Volume of right limb(cm³)** | 2492.5±326.1 | 2255.7±758.9 | 1586.4±287.1 | **0.001** |
| **Volume of left limb(cm³)** | 3505.7±337.9 | 2691.7±755.4 | 1630.01±316.3 | **0.001** |

Data expressed as mean±SD; *One way ANOVA test

In our study, most patients were diagnosed at stage III of the disease and developed lymphedema an average of 16.6±10 months after completing their last treatment. This finding aligns with a large-scale study reporting that breast cancer patients developed lymphedema within 12 months of surgery [17]. In contrast, an Iranian study reported a longer average duration of 37 months for the onset of lymphedema post-surgery. Additionally, another study found that patients who underwent both axillary dissection and radiation therapy had a higher risk of developing upper limb lymphedema

**Table 3.** Difference of mean volume change of lymphedema limbs over subsequent visits (n = 42).

| Pair of visits | | Mean difference | 95% CI | | P value* |
|---|---|---|---|---|---|
| | | | Lower bound | Upper bound | |
| Baseline visit | 1st follow up visit | 236.8 | 179.8 | 653.4 | 0.492 |
| Baseline visit | 2nd follow up visit | 906.1 | 489.5 | 1322.7 | **0.001** |
| 1st follow up visit | 2nd follow up visit | 669.3 | 252.7 | 1085.9 | **0.001** |

*bonferroni test*

**Table 4.** Comparison of skin conditions between different visits (n = 42).

| Variables | Baseline | 1st visit | 2nd visit | P value* |
|---|---|---|---|---|
| | Frequency (%) | | | |
| **Skin edema** | | | | |
| Pitting oedema | 2 (4.8) | 4 (9.5) | 4 (9.5) | 0.429 |
| Non-pitting oedema | 40 (95.2) | 38 (90.5) | 38 (90.5) | |
| **Skin texture** | | | | |
| Normal | 10 (23.8) | 18 (42.8) | 25 (59.5) | 0.301 |
| Dry and shiny | 32 (76.2) | 24 (57.2) | 17 (40.5) | |

*chi-square test was done*

**Table 5.** Comparison of self-reported symptoms and observed signs between different visits (n = 42).

| Variables | Baseline | 1st visit | 2nd visit | P value* |
|---|---|---|---|---|
| | Frequency (%) | | | |
| Tightness | 38 (90.5) | 15 (35.7) | 5 (11.9) | **0.021** |
| Heaviness | 39 (92.9) | 15 (35.7) | 5 (11.9) | **0.023** |
| Bursting feeling | 15 (35.7) | 19 (45.2) | 7 (16.6) | 0.456 |
| Pain | 18 (42.9) | 8 (19.0) | 2 (4.7) | **0.041** |
| Impaired function/mobility | 26 (61.9) | 23 (54.8) | 15 (35.7) | 0.892 |
| Problems in obtaining well-fitting clothes | 38 (90.5) | 29 (69.0) | 23 (54.8) | 0.975 |
| Stemmer's sign | 27 (64.3) | 18 (42.8) | 10 (23.8) | 0.276 |

*Chi-square test*

compared to those treated with either modality alone [18]. Our findings are consistent with this, as the majority of our patients received multiple treatment modalities for breast cancer.

All of our patients presented with moderate to severe upper limb lymphedema and received the intensive phase of Complex Decongestive Therapy (CDT). We observed significant limb volume reduction in both right and left upper limbs across follow-up visits, with an average decrease of 906.08 mL from the baseline to the 6-week follow-up. Sezgin Ozcan et al. similarly reported a reduction of 248.9 mL after 10 days of intensive CDT [19], while another study observed a median reduction of 328.7 mL following just 6 days of treatment [20]. Comparable outcomes have been documented in studies conducted in Poland, the USA, Brazil, Iran, and Ireland, where patients

receiving CDT demonstrated significant reductions in limb volume [16,21–24]. A retrospective study from the USA also found meaningful volume reduction in post-mastectomy patients following CDT, irrespective of prior lymphedema therapy [13].

However, in most of these studies, the intensive phase of CDT lasted between 6 days and 3 weeks. Only two studies involving palliative lymphedema patients reported extended treatment durations ranging from 12 weeks to 6 months [16,22]. While the mean volume reduction observed in our study was notably higher than in previous reports, this may be attributed to the longer treatment duration of 6 weeks. Therefore, our findings suggest that CDT alone is effective in reducing edema among palliative lymphedema patients, even in resource-limited settings such as Bangladesh.

Regarding skin changes following CDT, we observed a reduction in non-pitting edema over consecutive visits, whereas pitting edema remained relatively unchanged during the second and third visits. In terms of skin texture, improvements were noted in dryness and shininess, with nearly half of the patients regaining normal skin appearance by the second and third follow-up visits; however, these changes were not statistically significant. In contrast, a study conducted in Ireland reported significant improvements in both skin texture and edema reduction following CDT [16].

In our study, skin changes were assessed clinically due to the unavailability of validated assessment tools and advanced hospital facilities. Conversely, Lee et al. utilized ultrasonography to evaluate the effects of CDT in patients with breast cancer-related lymphedema (BCRL). Their findings demonstrated a significant reduction in soft tissue thickness at the elbow and at a point 10 cm proximal to the elbow during follow-up. This suggests that combining arm circumference measurements with ultrasonographic evaluation of soft tissue thickness may serve as effective methods for assessing CDT outcomes in BCRL patients [25].

Furthermore, we observed significant improvement in limb tightness, heaviness, and pain following the intensive phase of CDT. This aligns with findings from a Korean study, which reported notable improvements in physical function and bodily pain after CDT [26]. In that study, reductions in pain, heaviness, and impaired limb mobility were significant during Phase I and sustained during Phase II. Similarly, Liang et al. reported significant improvements in general health and vitality after three months of CDT [19,27]. Collectively, these findings underscore the effectiveness of CDT in alleviating both physical symptoms and discomfort in patients with upper limb lymphedema.

## Limitations

This study has several limitations. First, it was conducted on a small, non-random, purposive sample over a short duration, and we were unable to assess the effects during the maintenance phase of CDT. Second, as data were collected from a single center, the generalizability of the findings to a broader population is limited.

## Conclusion

Lymphedema management utilizing all components of Complex Decongestive Therapy (CDT) was found to be effective in reducing limb volume and alleviating distressing symptoms in patients. To sustain these improvements, continued application of CDT over time is essential. Longer-term follow-up studies are warranted to determine the optimal treatment duration and to assess the long-term efficacy of CDT. Timely referral of lymphedema patients to specialized clinics and early initiation of CDT can play a crucial role in reducing their ongoing suffering, particularly in resource-limited settings such as Bangladesh.

## Acknowledgments

The authors gratefully acknowledge the contribution of Dr. Sohel Rahman for his assistance in analyzing the data for this study.

## Author contributions

**Conceptualization:** Rifat Jahan, A.K.M. Motiur Rahman Bhuiyan, Afroja Alam, Mostofa Kamal Chowdhury.

**Data curation:** Rifat Jahan, Afroja Alam, Mostofa Kamal Chowdhury.

**Formal analysis:** Rifat Jahan, Afroja Alam.

**Funding acquisition:** Rifat Jahan, A.K.M. Motiur Rahman Bhuiyan, Afroja Alam, Mostofa Kamal Chowdhury.

**Investigation:** Rifat Jahan, Mostofa Kamal Chowdhury.

**Methodology:** Rifat Jahan, A.K.M. Motiur Rahman Bhuiyan, Afroja Alam, Mostofa Kamal Chowdhury.

**Project administration:** Rifat Jahan.

**Resources:** Rifat Jahan, A.K.M. Motiur Rahman Bhuiyan, Afroja Alam.

**Software:** Rifat Jahan, Afroja Alam.

**Supervision:** A.K.M. Motiur Rahman Bhuiyan, Afroja Alam, Mostofa Kamal Chowdhury.

**Validation:** Rifat Jahan, Afroja Alam, Mostofa Kamal Chowdhury.

**Visualization:** Rifat Jahan, A.K.M. Motiur Rahman Bhuiyan.

**Writing – original draft:** Rifat Jahan, A.K.M. Motiur Rahman Bhuiyan, Afroja Alam, Mostofa Kamal Chowdhury, Romena Hassan, Nisha Mortuja Aktar.

**Writing – review & editing:** Rifat Jahan, A.K.M. Motiur Rahman Bhuiyan, Afroja Alam, Mostofa Kamal Chowdhury, Romena Hassan, Nisha Mortuja Aktar.

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
