## [Decision Letter · Decision Letter 0]

Outcomes of Complex Decongestive Therapy in Managing Upper Limb Lymphedema in Female Breast Cancer Patients at a Palliative Care Unit of a Tertiary Care Hospital in Bangladesh

PLOS ONE

Dear Dr. Alam,

Thank you for submitting your manuscript to PLOS ONE. After careful consideration, we feel that it has merit but does not fully meet PLOS ONE’s publication criteria as it currently stands. Therefore, we invite you to submit a revised version of the manuscript that addresses the points raised during the review process.

We have invited more than 20 experts to review the ms. Of them only two responded. One of the reviewers raised several comments on the technical aspects of the study, which indicate that the methodology needs substantial revision of the ms. I would request the authors, in addition to the minor points, to focus on the methodological issues raised by Reviewer 2.

We look forward to receiving your revised manuscript.

Kind regards,

Swaminathan Subramanian, Ph.D.

Academic Editor

PLOS ONE

Journal Requirements:

Additional Editor Comments:

NIL

Reviewers' comments:

Reviewer's Responses to Questions

**Comments to the Author**

1. Is the manuscript technically sound, and do the data support the conclusions?

Reviewer #1: Yes

Reviewer #2: Partly

2. Has the statistical analysis been performed appropriately and rigorously?

Reviewer #1: Yes

Reviewer #2: Yes

3. Have the authors made all data underlying the findings in their manuscript fully available?

Reviewer #1: Yes

Reviewer #2: Yes

4. Is the manuscript presented in an intelligible fashion and written in standard English?

Reviewer #1: No

Reviewer #2: Yes

Reviewer #1: The aim of this study is to evaluate the effectiveness of CDT for breast cancer patients with upper limb

lymphedema and aims to assess the benefits of this treatment despite the challenges

and constraints in resource-limited settings. It is well structured in all parts. Only the language must be reviewed.

Reviewer #2: It is a well-written manuscript. However there are a few clarifications and suggestions which the authors can address for better clarity.

1. It is not clear in the methodology if the upper limb lymphedema (LE) is unilateral or bilateral. Based on the Results it can be presumed to be bilateral LE.

2. How LE was graded as moderate or severe can be mentioned in the Methodology

3. In line 122, it is mentioned that data collection was done between 1 May 2023 and 30 Sep 2023. Does it mean the recruitment was stopped or it implies that the final 6 weeks follow-up post treatment (CDT) was completed for all participants on 30 Sep 2023. Rephrase the sentence for better clarity.

4. Lines 162 -165: provide reference

5. Line 168: What is the interpretation for the colour domains mentioned: Normal, red, pink, brown?

6. Line 169: Skin thickness can be either normal, thin or thick (stemmer's sign can be mentioned). However, edema can be a separate assessment domain and needn't be mixed with skin thickness.

7. Line 177-178: Mention that the CDT was provided for 6 weeks

8. Lines 197, 200, 202: Ensure either one or two decimal points is provided uniformly in the text as well as in the tables (Table 1 Percentage% check the decimals)

9. Was there any association between stage of cancer and improvement of LE?

10. Table 2: Volume of left limb it is mentioned 3505.69+- 3307.87- please recheck value

11. Table 2: the baseline volume for left limb is observed to be greater compared to the right limb. Is it because left limb LE was more common among the participants?

12. Lines 222-223: Can improvement in skin condition be attributed to CDT?

13. Table 4: Skin thickness and edema can be two separate domains

14. Lines 262-264: The factors discussed were not assessed in the present study. Its relevance here is not clear

15. Line 267: Skin thickness and edema to be discussed separately.

16. Line 295: Lower limb LE was excluded and not relevant here

17. Lines 65 & 301:" early referral..."- the study did not record the delay in accessing treatment (CDT) by patients. Justify the statement provided here.

**Do you want your identity to be public for this peer review?** For information about this choice, including consent withdrawal, please see our Privacy Policy

Reviewer #1: No

Reviewer #2: No

---

## [Author Response · Author response to Decision Letter 1]

30 Apr 2025

Reviewer#1’s comments to the authors-

Comment: The aim of this study is to evaluate the effectiveness of CDT for breast cancer patients with upper limb lymphedema and aims to assess the benefits of this treatment despite the challenges and constraints in resource-limited settings. It is well structured in all parts. Only the language must be reviewed.

Reply: Thank you. We have revised the language for grammatical errors.

Reviewer#2 comments to the author-

Comment: It is not clear in the methodology if the upper limb lymphedema (LE) is unilateral or bilateral. Based on the Results it can be presumed to be bilateral LE.

Reply: Thank you. All patients had unilateral lymhedema. The number of affected limbs are mentioned in revised Table 1

Comment: How LE was graded as moderate or severe can be mentioned in the Methodology

Reply 2: Thank you. We have added the lymphdedema grading in the methods section.

Changes in the text:

Line 135-150: Lymphedema grading: The severity of lymphedema in the affected limb was categorized based on circumferential measurements in comparison to the contralateral, unaffected limb [12]-

• Grade 1 (Mild): Involves distal regions such as the forearm and hand or lower leg and foot. Circumferential difference is less than 4 cm, with no observable tissue changes.

• Grade 2 (Moderate): Affects the entire limb or the corresponding trunk quadrant, with a circumferential difference of 4–6 cm. Observable tissue alterations such as pitting may occur, and patients may present with erysipelas.

• Grade 3a (Severe): Lymphedema extends to one limb and its adjacent trunk quadrant, with a circumferential difference exceeding 6 cm. Skin changes including cornification, keratosis, cysts, or fistulae are evident, and recurrent episodes of erysipelas may be reported.

• Grade 3b (Massive): Characterized by the involvement of two or more extremities, exhibiting the same features as Grade 3a.

• Grade 4 (Gigantic): Marked by extreme enlargement of the affected limbs, attributed to near-total obstruction of lymphatic flow.

Comment: In line 122, it is mentioned that data collection was done between 1 May 2023 and 30 Sep 2023. Does it mean therecruitment was stopped or it implies that the final 6 weeks follow-up post treatment (CDT) was completed for allparticipants on 30 Sep 2023. Rephrase the sentence for better clarity.

Reply: Thank you. We revised the line.

Changes in the text:

Line 122-124: Participant recruitment commenced on 1 May 2023 and continued through 18 August 2023. The six-week post-treatment follow-up for all enrolled patients was concluded by 30 September 2023.

Comment: Lines 162 -165: provide reference

Reply: Thank you. The reference has been provided- Ref no 15 and 16

Comment: Line 168: What is the interpretation for the color domains mentioned: Normal, red, pink, brown?

Reply: Thank you. We have again checked the reference (ref 16) and based on the tool used there we limited the skin integrity domains into two (color change is deleted).

Comment: Line 169: Skin thickness can be either normal, thin or thick (stemmer's sign can be mentioned). However, edema can be a separate assessment domain and needn't be mixed with skin thickness.

Reply: Thank you. We have corrected the issue. Based on the tool used (ref 16), we added skin edema as separate domain. Skin thickness was actually assed separately, only stemmer’s sign was assed as part of clinical evaluation which was mention in table 5.

Comment: Line 177-178: Mention that the CDT was provided for 6 weeks

Reply: Thank you. We have corrected the line.

Changes in the text:

Line 191-194: All patients underwent intensive CDT phase 1 for 6 weeks during the study period. No alterations to the treatment protocol were made by the investigators. The patients were followed up after 3 weeks during the mid-procedure and after 6 weeks after competition of 6 weeks of CDT.

Comment: Lines 197, 200, 202: Ensure either one or two decimal points is provided uniformly in the text as well as in the tables (Table 1 Percentage% check the decimals)

Reply: Thank you. We have corrected the lines.

Comment: Was there any association between stage of cancer and improvement of LE?

Reply: We found no association between cancer stage and improvement of LE

Comment: Table 2: Volume of left limb it is mentioned 3505.69+- 3307.87- please recheck value

Reply: Thank you. We have corrected the value.

Comment: Table 2: the baseline volume for left limb is observed to be greater compared to the right limb. Is it because left limb LE was more common among the participants?

Reply: Our participant had mostly left limb LE. The number of affected limb is mentioned in revised Table 1. So the baseline volume for LE was higher in left limb in our study.

Comment: Lines 222-223: Can improvement in skin condition be attributed to CDT?

Reply: Yes it can be attributed to CDT.

Changes in the text:

Line 240-242: However, the improvement in skin condition can be attributed to CDT, as it not only resulted in volume reduction but also included proper skin care

Comment: Table 4: Skin thickness and edema can be two separate domains

Reply: Skin thickness was not separately measured. So we changed the heading as skin edema in table 4

Comment: Lines 262-264: The factors discussed were not assessed in the present study. Its relevance here is not clear

Reply: Thank you. As these factors are not assessed in our study, so we deleted the line.

Comment: Line 267: Skin thickness and edema to be discussed separately.

Reply: Thank you. We made the change accordingly.

Changes in the text:

Line 282-289: Regarding skin changes with CDT, we observed a decrease in non-pitting edema over consecutive visits, while pitting edema remained relatively unchanged on the 2nd and 3rd visits. Also regarding skin texture, In terms of skin texture, dryness and shiny skin showed improvement, with nearly half of the patients regaining normal skin by the second and third visits, although these changes were not statistically significant. In contrast, a study in Ireland reported significant improvement in skin texture and decrease in edema [16].

Comment: Line 295: Lower limb LE was excluded and not relevant here

Reply: Thank you. We deleted the line.

Comment: Lines 65 & 301:" early referral..."- the study did not record the delay in accessing treatment (CDT) by patients. Justifythe statement provided here.

Reply: Thank you. We made the change accordingly

Changes in the text:

Line 65 and 313: Referral of lymphedema patients to specialized clinics and the timely initiation of CDT can significantly reduce ongoing suffering and contribute to overall improvements in quality of life for patients in Bangladesh

Dear Reviewers, we are grateful for your kind time and substantial review; we believe now the manuscript is more improved, which will satisfy you.

---

## [Decision Letter · Decision Letter 1]

Dear Dr. Alam,

Thank you for submitting your manuscript to PLOS ONE. After careful consideration, we feel that it has merit but does not fully meet PLOS ONE’s publication criteria as it currently stands. Therefore, we invite you to submit a revised version of the manuscript that addresses the points raised during the review process.

Both reviewers recommended for a minor revision. However, one of the reviewers raised several minor comments which needs to be addressed. For example the data provided in Tables 3 and 4 do not match with Results presented. Also make appropriate changes as indicated in the attached Comments to Authors. 

We look forward to receiving your revised manuscript.

Kind regards,

Swaminathan Subramanian, Ph.D.

Academic Editor

PLOS ONE

Journal Requirements:

Reviewers' comments:

Reviewer's Responses to Questions

**Comments to the Author**

Reviewer #1: All comments have been addressed

Reviewer #2: (No Response)

2. Is the manuscript technically sound, and do the data support the conclusions?

Reviewer #1: Yes

Reviewer #2: Yes

3. Has the statistical analysis been performed appropriately and rigorously?

Reviewer #1: Yes

Reviewer #2: Yes

4. Have the authors made all data underlying the findings in their manuscript fully available?

Reviewer #1: Yes

Reviewer #2: Yes

5. Is the manuscript presented in an intelligible fashion and written in standard English?

Reviewer #1: No

Reviewer #2: No

Reviewer #1: This study seeks to evaluate the effectiveness of CDT for breast cancer patients with upper limb

lymphedema and aims to assess the benefits of this treatment despite the challenges

and constraints in resource-limited settings. The paper is well structured but the English must be improved.

Reviewer #2: The authors have addressed most of the comments that were raised during the previous review. However, there are few more additional comments that needs to be addressed by them:

1. The authors responded that all 42 patients had unilateral (U/L) upper limb lymphedema (LE). Whether U/L LE was one of the inclusion criteria? If so, it may be mentioned in the 'Methods' section in both Abstract and Main text of the manuscript (MS).

2. Line 56: "Significant volume reduction in both right and left upper limbs.." This is quite misleading as all cases had unilateral LE. The sentence shall be reframed for clarity.

3. Lines 61-62 and 67, 314: Tightness, heaviness and pain are more subjective responses. It is understood that no validated scale to assess quality of life (QoL) was used in the study and hence the term QoL may be avoided.

4. Line 125: Correct the spelling of 'massive'

5. Line 135: Ref 12 has used LE grading by International Society of Lymphology (ISL), the same shall be mentioned in the MS. These grades are found nowhere else other than the Methods. Information on classification of study participants based on these grades needs to be included in the MS. Were these LE grades not considered for the analysis?

6. Line 172: Please correct the tense (Limb volume changes were...)

7. Line 173-174: '...using a measuring tape along with the fixed designated point...' It is not clear. Rephrase the sentence for clarity

8. Table 1: Write Age (years) with Mean (SD): 50.2+-8.6 and remove years in the rows.

9. Table 1: Performance status- Correct the decimal points (one decimal) for uniformity. Stage I - III is used which is confusing. Please mention it as the Stage of breast cancer for clarity. Further there is no mention of these stages in the Methods.

10. Table 2: Correct decimal points for the volume mentioned (2492.4981, 2691.70 etc...)

11. Line 230-231: Table 3 information cited doesn't match with the sentence provided.

12. Table 3, 4: Title and information provided in tables do not match

13. Table 3: Ensure uniform font size

14. Lines 236-243: Table 4 is cited wrongly in the text

15. Line 248: Tightness, heaviness and pain...these are more subjective symptoms and supposedly assessed orally (using questionnaire). It may be clarified that these symptoms were self-reported perceptions by the patients.

16. Line 258: Check for grammatical error

17. Line 270: Correct the decimal points

18. Previous comment: The authors have replied that no association was found between stage of cancer and improvement of LE. The same shall be mentioned in the Results. Also was there any association found between the LE grades and improvement in LE? If so, please mention in the MS.

Kindly check the MS for grammatical errors once again

**Do you want your identity to be public for this peer review?** For information about this choice, including consent withdrawal, please see our Privacy Policy

Reviewer #1: No

Reviewer #2: No

---

## [Author Response · Author response to Decision Letter 2]

12 May 2025

Outcomes of Complex Decongestive Therapy in Managing Upper Limb Lymphedema in Female Breast Cancer Patients at a Palliative Care Unit of a Tertiary Care Hospital in Bangladesh

(Manuscript ID: PONE-D-24-46999.R1)

Reviewer#1’s comments to the authors-

Comment: This study seeks to evaluate the effectiveness of CDT for breast cancer patients with upper limb lymphedema and aims to assess the benefits of this treatment despite the challenges and constraints in resource-limited settings. The paper is well structured but the English must be improved.

Reply: Thank you. We have revised the language for grammatical errors.

Reviewer#2 comments to the author-

Comment: 1. The authors responded that all 42 patients had unilateral (U/L) upper limb lymphedema (LE). Whether U/L LE was one of the inclusion criteria? If so, it may be mentioned in the 'Methods' section in both Abstract and Main text of the manuscript (MS)..

Reply: Thank you. We have updated the inclusion criteria.

Comment: 2. Line 56: "Significant volume reduction in both right and left upper limbs.." This is quite misleading as all cases had unilateral LE. The sentence shall be reframed for clarity.

Reply 2: Thank you. We have corrected the line.

Changes in the text:

Line 56: A significant reduction in the volume of the affected limbs was observed from baseline to the 6th week, as well as from the 3rd week to the 6th week]-

Comment: 3. Lines 61-62 and 67, 314: Tightness, heaviness and pain are more subjective responses. It is understood that no

validated scale to assess quality of life (QoL) was used in the study and hence the term QoL may be avoided..

Reply: Thank you. We revised the line.

Changes in the text:

Line 65, 324: Timely referral of lymphedema patients to specialized clinics and initiation of CDT can significantly reduce their ongoing suffering in Bangladesh.

Comment: 4. Line 125: Correct the spelling of 'massive'

Reply: Thank you. The spelling was corrected.

Comment: Line 135: Ref 12 has used LE grading by International Society of Lymphology (ISL), the same shall be mentioned in the MS. These grades are found nowhere else other than the Methods. Information on classification of study participants based on these grades needs to be included in the MS. Were these LE grades not considered for the analysis?

Reply: Thank you. We updated the reference (ref 12) and added the no of pt according to the grading (table 1). However this grading was only used as inclusion criteria, but not used in analysis.

Comment: 6. Line 172: Please correct the tense (Limb volume changes were...).

Reply: Thank you. We have corrected the line.

Comment: Line 173-174: '...using a measuring tape along with the fixed designated point...' It is not clear. Rephrase the sentence for clarity

Reply: Thank you. We have corrected the line.

Changes in the text:

Line 172: Limb volume changes were measured by assessing the circumference of the affected limb using a measuring tape at fixed designated points, and the limb volume was then calculated using the truncated cone formula.

Comment: Table 1: Write Age (years) with Mean (SD): 50.2+-8.6 and remove years in the rows.)

Reply: Thank you. We have made the correction according to your suggestion.

Comment: Table 1: Performance status- Correct the decimal points (one decimal) for uniformity. Stage I - III is used which is confusing. Please mention it as the Stage of breast cancer for clarity. Further there is no mention of these stages in the Methods.

Reply: We made the correction according to your suggestion (table 1).

Comment: Table 2: Correct decimal points for the volume mentioned (2492.4981, 2691.70 etc...)

Reply: Thank you. We have corrected the values.

Comment: Line 230-231: Table 3 information cited doesn't match with the sentence provided.

Reply: correction was made accordingly.

Comment: Table 3, 4: Title and information provided in tables do not match

Reply: Table titles are corrected (table 3, 4)

Comment: Table 3: Ensure uniform font size

Reply: Text size revised.

Comment: Lines 236-243: Table 4 is cited wrongly in the text

Reply: Thank you. We made the suggested corrections.

Comment: Tightness, heaviness and pain...these are more subjective symptoms and supposedly assessed orally(using questionnaire). It may be clarified that these symptoms were self-reported perceptions by the patients.

Reply: Thank you. We made added ‘self-reported symptoms and observed signs’ in the title of table 5.

Comment: Line 258: Check for grammatical error

Reply: Thank you. The line has been checked and corrected.

Comment: Correct the decimal points.

Reply: Thank you. Suggested corrections are made.

Comment: Previous comment: The authors have replied that no association was found between stage of cancer and improvement of LE. The same shall be mentioned in the Results. Also was there any association found between the LE grades and improvement in LE? If so, please mention in the MS..

Reply: Thank you. Suggested lines are mentioned in the result. As we used lymphedema grading only as inclusion criteria, but didn’t included in the analysis, so we didn’t actually assessed the change in lymphedema grade with volume change

Changes in the text:

Line 172: However, we did not find any association between cancer stage and the volume reduction of the affected limb

Dear Reviewers, we are grateful for your kind time and substantial review; we believe now the manuscript is more improved, which will satisfy you.

---

## [Decision Letter · Decision Letter 2]

Outcomes of Complex Decongestive Therapy in Managing Upper Limb Lymphedema in Female Breast Cancer Patients at a Palliative Care Unit of a Tertiary Care Hospital in Bangladesh

PONE-D-24-46999R2

Dear Dr. Alam,

We’re pleased to inform you that your manuscript has been judged scientifically suitable for publication and will be formally accepted for publication once it meets all outstanding technical requirements.

Kind regards,

Swaminathan Subramanian, Ph.D.

Academic Editor

PLOS ONE

Additional Editor Comments (optional):

I would request the authors tio revise the ms incorporating the suggestions against comment no. 8 (on Revision 1) and Table 1 (please see comments of Reviewer 2 below for more detail)

Reviewers' comments:

Reviewer's Responses to Questions

**Comments to the Author**

Reviewer #2: All comments have been addressed

2. Is the manuscript technically sound, and do the data support the conclusions?

Reviewer #2: (No Response)

3. Has the statistical analysis been performed appropriately and rigorously?

Reviewer #2: (No Response)

4. Have the authors made all data underlying the findings in their manuscript fully available?

Reviewer #2: (No Response)

5. Is the manuscript presented in an intelligible fashion and written in standard English?

Reviewer #2: (No Response)

Reviewer #2: The authors have addressed all the comments. However, with regard to comment no:8, authors have wrongly interpreted and deleted the age classes in Table 1 along with relevant text in results section. This may be reincluded.

In Table 1, write as follows:

Age class in years (Mean±SD 50.2±8.6)

Min age-49 10 45.2%

50-Max age 23 54.8%

**Do you want your identity to be public for this peer review?** For information about this choice, including consent withdrawal, please see our Privacy Policy

Reviewer #2: No

---

## [Editor Report · Acceptance letter]

PONE-D-24-46999R2

PLOS ONE

Dear Dr. Alam,

I'm pleased to inform you that your manuscript has been deemed suitable for publication in PLOS ONE. Congratulations! Your manuscript is now being handed over to our production team.

Kind regards,

on behalf of

Dr. Swaminathan Subramanian

Academic Editor

PLOS ONE